

# A new species of *Leptobrachium* (Anura, Megophryidae) from western Thailand

Parinya Pawangkhanant[1], Nikolay A. Poyarkov[2,3], Tang Van Duong[2,4], Mali Naiduangchan[5] and Chatmongkon Suwannapoom[1]

[1] Division of Fishery, School of Agriculture and Natural Resources, University of Phayao, Phayao, Thailand
[2] Faculty of Biology, Department of Vertebrate Zoology, Moscow State University, Moscow, Russia
[3] Laboratory of Tropical Ecology, Joint Russian-Vietnamese Tropical Research and Technological Center, Hanoi, Vietnam
[4] Vietnam National Museum of Nature, Vietnam Academy of Science and Technology, Hanoi, Vietnam
[5] Rabbit in the Moon Foundation, Suan Phung, Ratchaburi, Thailand

Corresponding authors
Nikolay A. Poyarkov,
n.poyarkov@gmail.com
Chatmongkon Suwannapoom,
chatmongkonup@gmail.com

## ABSTRACT

We describe a new species of the genus *Leptobrachium* from the Khao Laem Mountain, Suan Phung District, Ratchaburi Province, Tenasserim Region, western Thailand, based on molecular and morphological evidences. The new species, *Leptobrachium tenasserimense* sp. nov., can be distinguished from all other congeners by the following combination of characters: (1) adult SVL of 41.4–58.8 mm in males and 54.7–58.6 mm in females; (2) rounded finger and toe tips; (3) relative finger lengths: II<IV<I<III; relative toe lengths: I<II<V<III<IV; (4) toe webbing thick and well developed; (5) inner metatarsal tubercle small; (6) iris bicolored, black ventrally and turquoise dorsally, with light blue sclera; (7) dorsum brown to grey with distinct darker markings edged with brown; (8) belly and limbs ventrally whitish with contrasting confluent black reticulations; (9) tympanum mostly free of dark marking; (10) narrow dark canthal stripe present; (11) lateral row of dark spots absent; (12) limbs dorsally with distinct dark bars; tibia with four to five dark transverse bars; (13) dense dark reticulation or large dark blotch at groin continuing to ventral and posterior sides of thighs; (14) femoral gland in shape of large white blotch; (15) males with single vocal sac, mature males lack lip spinules. Our study provides further evidence for a hidden biodiversity of montane areas of Tenasserim Region on the border of Thailand and Myanmar.

## INTRODUCTION

The frog family Megophryidae is a key element of Southeast Asian herpetofauna, currently includes 218 species, and is distributed throughout Pakistan and southern China southwards to the Philippines and the Greater Sunda Islands (*Frost, 2018*). Due to the high unrecognized diversity and morphological similarity of many species within the family, molecular phylogenetic tools are crucial for studies of the group's taxonomy, which currently remains in a constant state of flux (*Matsui et al., 2010*; *Matsui, 2013*; *Poyarkov et al., 2015*; *Poyarkov et al., 2017*; *Chen et al., 2017*; *Chen et al., 2018*).

The genus *Leptobrachium* Tschudi, 1838 (Asian spadefoot toads) currently includes 35 species, which are widely distributed from southern China westwards to northeastern India and Myanmar, through Indochina mainland to peninsular Malaysia, Borneo, Sumatra, Java and Philippines (*Frost, 2018*; *Sondhi & Ohler, 2011*; *Stuart et al., 2011*; *Stuart et al., 2012*; *Yang, Wang & Chan, 2016*). *Leptobrachium* frogs have recently been included in several phylogenetic studies (*Frost et al., 2006*; *Rao & Wilkinson, 2008*; *Zheng, Li & Fu, 2008*; *Brown et al., 2009*; *Zhang et al., 2010*; *Matsui et al., 2010*; *Wogan, 2012*; *Yang, Wang & Chan, 2016*; *Li et al., 2018*). At the genus level, all these studies agree that *Leptobrachium* species are grouped into two major clades: the Sundaland/western Indochina Clade, composed of species from Sundaland, southern Thailand and southern Myanmar, and the China/eastern Indochina Clade, composed of species from eastern Indochina and southern China, including Hainan Island. Though several studies addressed phylogenetic relationships within *Leptobrachium*, only in work of *Matsui et al. (2010)* these two clades were regarded as distinct subgenera, with Sundaland/western Indochina Clade corresponding to the subgenus *Leptobrachium* sensu stricto, while China/eastern Indochina Clade corresponding to the subgenus *Vibrissaphora*. Most other works assigned to *Vibrissaphora* only large montane species with cornified spines developed in males during the breeding season (*Orlov, 2005*; *Yang, Wang & Chan, 2016*), which do not form a clade and are deeply nested within the China/eastern Indochina radiation. In the present paper we follow the subgeneric taxonomy of *Matsui et al. (2010)*. At the species level, a number of analyses have revealed the presence of unrecognized cryptic diversity with a number of previously unknown mitochondrial DNA (mtDNA) lineages found throughout Asia, likely corresponding to yet undescribed species (*Brown et al., 2009*; *Hamidy et al., 2012*; *Matsui et al., 2010*; *Yang, Wang & Chan, 2016*).

To date, three *Leptobrachium* species were documented in Thailand, which belong to two *Leptobrachium* subgenera: *Leptobrachium (Vibrissaphora) chapaense* (Bourret), however according to recent molecular data, populations from northern Thailand likely correspond to *L. huashen* Fei & Ye, described from the southern part of Yunnan Province of China (see *Yang, Wang & Chan, 2016* for discussion); *L. hendricksoni* Taylor; and *L. smithi* Matsui, Nabhitabhata & Panha, the latter two species belong to the subgenus *Leptobrachium* sensu stricto. Previous records of *L. hasseltii* Tschudi, *L. nigrops* Berry & Hendrickson, and *L. pullum* (Smith) for Thailand appear to be based on misidentifications or require further confirmation (see *Taylor, 1962*; *Matsui, Nabhitabhata & Panha, 1999*; *Wogan, 2012*; *Brown et al., 2009*). *Leptobrachium smithi* is a widespread species in western Indochina with red and black bicolored iris, historically confused with *L. pullum* described from southern Vietnam. *Matsui, Nabhitabhata & Panha (1999)* demonstrated its distinctiveness from *L. pullum* and described as a new species; consequent phylogenetic analysis demonstrated that *L. pullum* belongs to the different subgenus *Leptobrachium* sensu stricto, while *L. pullum* was assigned to the subgenus *Vibrissaphora* (*Matsui et al., 2010*). The same molecular study by *Matsui et al. (2010)* indicated that *L. smithi* populations in western Indochina consist of three highly divergent mtDNA lineages, forming a clade (see Fig. 1): (1) most populations of *L. smithi* sensu stricto from western Thailand, Langkawi Island in northern Malaysia and easternmost Myanmar; (2) populations from Rakhine State of Myanmar, later described

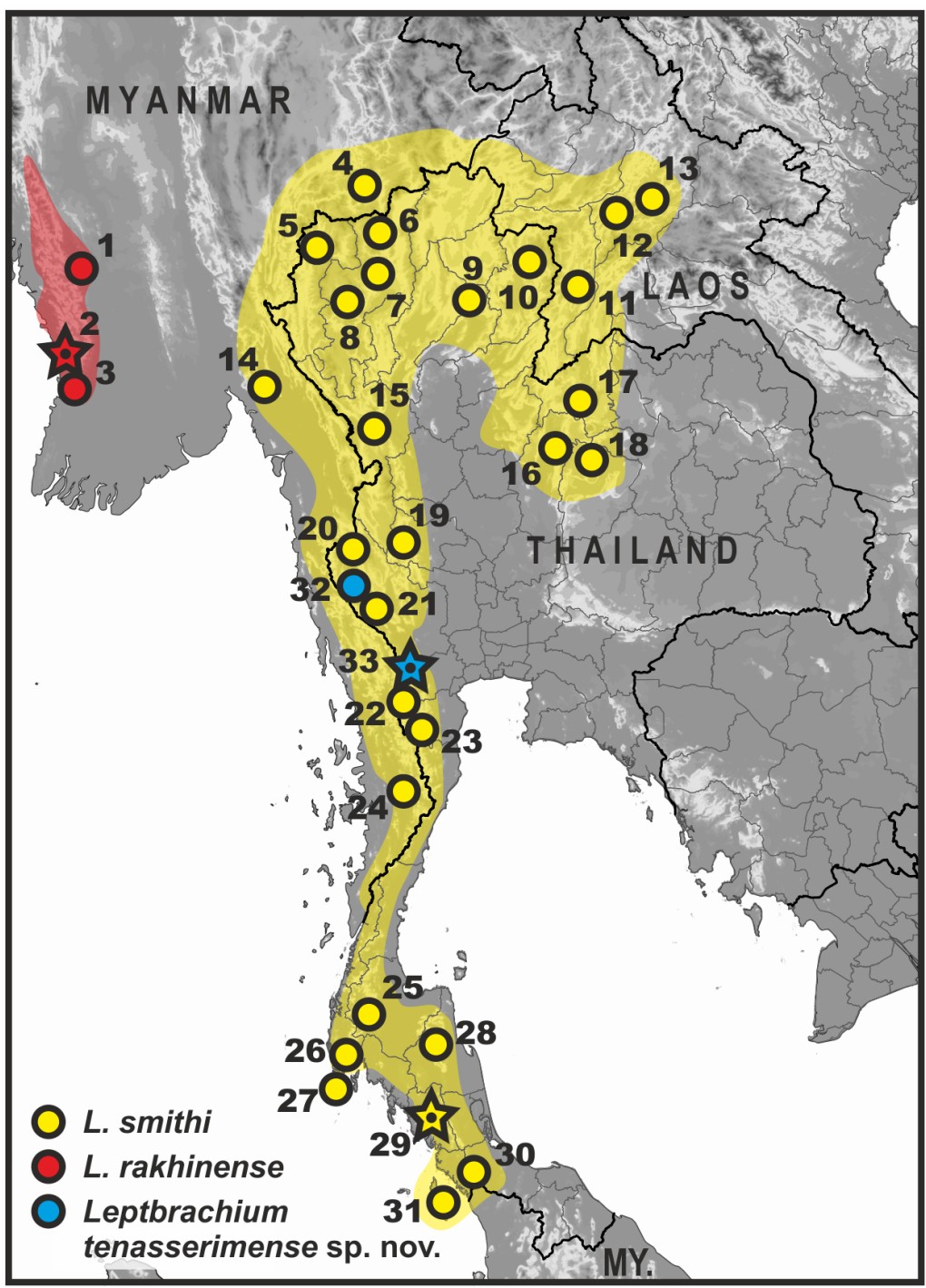

**Figure 1** **Map of Thailand and adjacent parts of Indochina, showing distribution of *L. smithi* species group members (clade L1).** Yellow, *L. smithi*; red, *L. rakhinense*; blue, *Leptobrachium tenasserimense* sp. nov. Star denotes type locality of the respective species. Locality information abbreviations: Distr., District; Div., Division; F.P., Forest Park; Isl., Island; N.P., National Park; Prov., Province; Res., Reserve; St., State; Twn., Township; W.F., waterfall; (continued on next page...)

**Figure 1 (...continued)**
W.S., Wildlife Sanctuary. *Leptobrachium rakhinensis* *Wogan, 2012*: 1-Nyaung Gwo, Padaung Twn., Pyi Distr., Bago Div., Myanmar (*Wogan, 2012*); 2-Rakhine Yoma W.S., Gwa Twn., Rakhine St., Myanmar (type locality) (*Wogan, 2012*); 3-Khoko Gwe, Rakhine Yoma W.S., Gwa Twn., Rakhine St., Myanmar (type locality) (*Wogan, 2012*). *Leptobrachium smithi* *Matsui, Nabhitabhata & Panha, 1999*: 4-Ma Gawe Res., Kalaw Twn., Taunggyi Dist., Shan St., Myanmar (*Wogan, 2012*); 5-Phasua W.F., Mae Hong Son Prov., Thailand (*Matsui, Nabhitabhata & Panha, 1999*); 6-Doi Chiang Dao Mt., Chiang Mai Prov., Thailand; 7-Doi Suthep Mt., Chiang Mai Prov., Thailand (*Matsui, Nabhitabhata & Panha, 1999*) ; 8-Doi Inthanon Mt., Chiang Mai Prov., Thailand (*Matsui, Nabhitabhata & Panha, 1999*); 9-Mae Yom N.P., Phrae Prov., Thailand (C Suwannapoom, 2018, unpublished data); 10-Tambol Auan, Amphoe Pua, Nan Prov., Thailand (FMNH 270740); 11-Houay Deng, Xaignabouli, Sayaboury Prov., Laos (*Brown et al., 2009*); 12-Houey Thao, Luang Prabang, Luang Prabang Prov., Laos (*Ohler et al., 2011*); 13-Ban Sop Khao, Ban Keng Koung, Ban Van Thong, Luang Prabang Prov., Laos (*Ohler et al., 2011*); 14-Kyaik Hti Yo W.S., Kyaihto Twn., Mon St., Myanmar (*Wogan, 2012*; *Matsui et al., 2010*); 15-Taksinmaharat N.P., Tak Prov., Thailand (P Pawangkhanant, 2018, unpublished data); 16-Thung Salaeng Luang N.P., Phetchabun Prov., Thailand (*Grosjean et al., 2015*); 17-Phu Luang N.P., Loei Prov., Thailand (*Matsui, Nabhitabhata & Panha, 1999*; *Matsui et al., 2010*); 18-Nam Nao N.P., Chaiyaphum Prov., Thailand (P Pawangkhanant, 2018, unpublished data); 19-Huai Kha Khaeng W.S., Uthai Thani Prov., Thailand (*Niyomwan, Srisom & Pawangkhanant, 2016*); 20-Sangkhla Buri Distr., Kanchanaburi Prov., Thailand (*Matsui, Nabhitabhata & Panha, 1999*); 21-Erawan and Pilok Distr., Kanchanaburi Prov., Thailand (*Matsui, Nabhitabhata & Panha, 1999*); 22-Kaeng Krachan, Phetchaburi Prov., Thailand (*Matsui et al., 2010*); 23-Pa Lao U, Phetchaburi Prov., Thailand (*Matsui, Nabhitabhata & Panha, 1999*); 24-Tanintharyi N.R., Yebyu Twn., Dawei Distr., Tanintharyi Div., Myanmar (*Wogan, 2012*); 25-Khlong Saen, Surat Thani Prov., Thailand (*Matsui, Nabhitabhata & Panha, 1999*); 26-Namtok Raman F.P.; Phang Nga Prov., Thailand (*Ohler et al., 2011*; *Grosjean et al., 2015*); 27-Phuket Isl., Phang Nga Prov., Thailand (*Matsui, Nabhitabhata & Panha, 1999*); 28-Khao Luang N.P., Nakhon Si Thammarat Prov., Thailand (*Matsui, Nabhitabhata & Panha, 1999*); 29-Kaochong, Trang Prov., Thailand (type locality) (*Matsui, Nabhitabhata & Panha, 1999*; *Matsui et al., 2010*); 30-Tha Le Ban National Park, Satun Prov., Thailand (P Pawangkhanant, 2018, unpublished data); 31-Langkawi Isl., Perlis, Malaysia (*Matsui, Nabhitabhata & Panha, 1999*; *Matsui et al., 2010*; *Grismer et al., 2006*). *Leptobrachium tenasserimense* **sp. nov.:** 32-Pilok Distr., Kanchanaburi Prov., Thailand (*Matsui et al., 2010*); 33-Khao Laem, Suan Phung Distr., Ratchaburi Prov., Thailand (type locality; sympatric with *L. smithi*) (this work).

as *L. rakhinensis* Wogan (due to the neutral gender of generic name, the species name has to be corrected to "*Leptobrachium rakhinense*"; see *Wogan, 2012*); (3) and a lineage recovered based on the analysis of tissues obtained from larval specimens of *Leptobrachium* sp., collected in Pilok District of Kanchanaburi Province in Tenasserim Region, western Thailand, originally denoted as "*Leptobrachium* sp. 4". In the present paper, we refer to these three mtDNA lineages as to the members of the *L. smithi* species complex.

During our field work in mountain areas of western Thailand we discovered and collected a medium-sized species of *Leptobrachium* that most closely resembles *L. smithi* and *L. rakhinense*, but distinctly differs from these and all other recognized congeners in morphological characters. Subsequent analyses of 16S rRNA mtDNA gene sequences confirmed that this population is conspecific with the lineage "*Leptobrachium* sp. 4" of *Matsui et al. (2010)* and confirmed that it represents an undescribed species of *Leptobrachium*. In this paper we present an updated mtDNA-based genealogy for *Leptobrachium*, and based on genetic and morphological comparisons, describe the Tennasserim population as a new species.

## MATERIAL AND METHODS

### Nomenclatural acts

The electronic version of this article in Portable Document Format (PDF) will represent a published work according to the International Commission on Zoological Nomenclature (ICZN), and hence the new names contained in the electronic version are effectively published under that Code from the electronic edition alone. This published work and the nomenclatural acts it contains have been registered in ZooBank, the online registration system for the ICZN. The ZooBank LSIDs (Life Science Identifiers) can be resolved and the associated information can be viewed through any standard web browser by appending the LSID to the prefix http://zoobank.org/. The LSID for this publication is as follows: urn:lsid:zoobank.org:pub:766A1EC7-4D17-412D-9DA6-C64E75EA3AFE. The online version of this work is archived and available from the following digital repositories: PeerJ, PubMed Central and CLOCKSS.

### Sampling

Specimens were collected on Khao Laem Mountain, Suan Phung District, Ratchaburi Province, western Thailand, by Parinya Pawangkhanant and Chatmongkon Suwannapoom. The geographic position of the surveyed locality and the distribution of the members of the *L. smithi* complex in Indochina are shown in Fig. 1. Geographic coordinates and elevation were obtained using a Garmin GPSMAP 60CSx and recorded in WGS 84 datum. Specimens were fixed in 10% buffered formalin after tissues were preserved in 95% ethanol. Specimens were later transferred to 70% ethanol. Specimens and tissues were subsequently deposited in the herpetological collections of the School of Agriculture and Natural Resources, University of Phayao (AUP, Phayao, Thailand) and of the Zoological Museum of Moscow University (ZMMU, Moscow, Russia).

Specimens collection protocols and animal use were approved by the Institutional Ethical Committee of Animal Experimentation of the University of Phayao, Phayao, Thailand (certificate number UP-AE59-01-04-0022 issued to Chatmongkon Suwannapoom) and were strictly complacent with the ethical conditions of the Thailand Animal Welfare Act. Field work, including collection of animals in the field and specimen exportation, was authorized by the Institute of Animals for Scientific Purpose Development (IAD), Bangkok, Thailand (permit number U1-01205-2558, issued to Chatmongkon Suwannapoom).

### Morphology

Sex of adult individuals was determined using gonadal dissection or by direct observation of calling in life. Measurements were taken to the nearest 0.01 mm using a digital caliper and subsequently rounded to a precision of 0.1 mm. The following 24 morphological characteristics were measured following *Matsui (1984)*: (1) snout-vent length (SVL); (2) head length (HL); (3) snout length (SL); (4) snout-nostril length (S-NL); (5) nostril-eyelid length (N-EL); (6) eye length (EL); (7) tympanum-eye length (T-EL); (8) tympanum diameter (TD); (9) head width (HW); (10) intercanthal distance (ICD); (11) internarial distance (IND); (12) interorbital distance (IOD); (13) upper eyelid width (UEW); (14) upper eyelid margin distance (UEMD); (15) fourth toe length (FTL); (16) first finger

length (FFL); (17) outer palmar tubercle length (OPTL); (18) inner palmar tubercle length (IPTL); (19) tibia length (TL); (20) foot length (FL); (21) hindlimb length (HLL); (22) hand length (HAL); (23) forearm width (FAW); and (24) inner metatarsal tubercle length (IMTL). Webbing formula is given following *Savage (1975)*. Terminology for eye coloration description in living individuals is in accordance with *Glaw & Vences (1997)*. Sexual size dimorphism index (SDI) was calculated following *Wogan (2012)* as mean female SVL divided by mean male SVL of adult individuals.

The morphological characteristics for comparison and the data on their states in other species of *Leptobrachium* were taken from the following studies: *Bain, Nguyen & Doan (2009)*, *Bourret (1937)*, *Chen et al. (2013)*, *Fei & Ye (2005)*, *Fei et al. (2009)*, *Fei, Ye & Jiang (2012)*, *Lathrop et al. (1998)*, *Matsui (2013)*, *Matsui, Nabhitabhata & Panha (1999)*, *Ohler, Teynié & David (2004)*, *Orlov (2005)*; *Pham et al. (2016)*, *Rao, Wilkinson & Zhang (2006)*, *Sondhi & Ohler (2011)*, *Smith (1921)*, *Stuart, Sok & Neang (2006)*, *Stuart et al. (2011)*, *Wogan (2012)*, *Yang, Wang & Chan (2016)*.

## DNA isolation and sequencing

Total DNA was extracted from ethanol-preserved muscle or liver tissues using standard phenol–chloroform extraction procedures (*Hillis, Moritz & Mable, 1996*) followed with isopropanol precipitation. We used the primers *16SL-1* and *16SH-1* from *Hedges (1994)* to amplify ~537 base pairs of the 16S rRNA mtDNA gene for the new species. For PCR conditions and primer sequences see *Poyarkov et al. (2015)*, *Nguyen et al. (2018)* and *Duong et al. (2018)*. PCR products were sent to Evrogen (Moscow, Russia) for subsequent purification and sequencing in both directions. The obtained sequences were checked by eye using Chromatogram Editor software DNABaser v4.20.0; primer sequences were removed and the edited sequences were submitted to GenBank under the accession numbers MH581080–MH581082 (Table S1).

## Phylogenetic analyses

The matrilineal genealogy was assumed to reflect the phylogenetic relationships of the species. To assess the genealogical relationships within the genus *Leptobrachium*, in addition to the newly collected specimens, 12S rRNA–16S rRNA mtDNA fragment sequences of all currently recognized *Leptobrachium* species were included in the genetic analysis (see Table S1 for details). Sequences of *Oreolalax rhodostigmatus* Hu & Fei, *Scutiger chintingensis* Liu & Hu, *Leptobrachella melanoleuca* (Matsui), *Megophrys nasuta* (Schlegel) (Megophryidae) and *Pelodytes punctatus* (Daudin) (Pelodytidae) were used as outgroups (Table S1). Sequences were initially aligned using the ClustalW (*Thompson et al., 1997*) and consequently checked and adjusted in Bioedit 7.0.5 (*Hall, 1999*) with default parameters. In total, a dataset of 81 ingroup and five outgroup sequences with a total length of up to 2494 bp was used for the analysis.

PartitionFinder v.1.1.0 (*Lanfear et al., 2012*) was applied to estimate the optimal evolutionary models for the dataset analysis. The best-fitting model of DNA evolution was the GTR+I+G, as suggested by the Akaike Information Criterion (AIC) and the Bayesian Information Criterion (BIC). The matrilineal genealogy was inferred using

Maximum Likelihood (ML) and Bayesian inference (BI) approaches. Maximum likelihood analysis was conducted in RAxML v8.2.4 (*Stamatakis, 2014*). BI analyses were conducted using MrBayes v.3.1.2 (*Huelsenbeck & Ronquist, 2001*; *Ronquist & Huelsenbeck, 2003*); Metropolis-coupled Markov chain Monte Carlo (MCMCMC) analyses were run with one cold chain and three heated chains for ten million generations and sampled every 1,000 generations; five independent MCMCMC runs were performed and 1,000 trees were discarded as burn-in. Confidence in node topology was assessed by posterior probability (BI PP, see (*Huelsenbeck & Ronquist, 2001*) for BI and by non-parametric bootstrapping with 1,000 replicates (ML BS, see *Felsenstein, 1985*) for ML analyses. Tree nodes with bootstrap (ML BS) values 70% or greater and Bayesian posterior probabilities (BI PP) values over 0.95 were *a priori* regarded as strongly supported (*Felsenstein, 2004*; *Hillis & Bull, 1993*; *Huelsenbeck & Hillis, 1993*), while ML BS values between 70% and 50% (BI PP between 0.95 and 0.90) were treated as tendencies and nodes with ML BS values below 50% (BI PP below 0.90) were regarded as not supported. Mean uncorrected genetic distances (*p*-distances) between sequences and species were calculated using MEGA 7.0 (*Kumar, Stecher & Tamura, 2016*).

## RESULTS

### Molecular relationships

Both BI and ML analyses resulted in highly similar topologies (Fig. 2). Genealogical relationships of the genus *Leptobrachium* correspond well to those reported by *Matsui et al. (2010)* and suggest its division into two reciprocally monophyletic groups, which correspond to the subgenus *Leptobrachium* sensu stricto from Sundaland, Philippines, Malayan Peninsula, western Thailand and Myanmar, and the subgenus *Vibrissaphora*, including species from southern China, northern Thailand and eastern Indochina (Vietnam, Cambodia and Laos) (see Fig. 2). The latter subgenus is subdivided into two subclades: V-I, comprised of species from southern China and northern Indochina, and subclade V-II, which includes species from southern and central parts of Annamite Mountains in eastern Indochina (Fig. 2). Subgenus *Leptobrachium* sensu stricto is subdivided into three subclades, the genealogical relationships among which are essentially unresolved: subclade L-I includes *L. smithi*, *L. rakhinense* and an undescribed species *Leptobrachium* sp. from Tenasserim Region of Thailand (*L. smithi* complex); subclade L-II comprises taxa from Borneo, Sumatra and Philippines (*L. montanum* Fischer complex); and subclade L-III includes species from southern Thailand, Malay Peninsula, Java, Bali, Sumatra and Borneo (*L. hasseltii–L. hendricksoni–L. nigrops* complex).

The unknown *Leptobrachium* species from the Ratchaburi Province of Thailand is predictably placed within the Sundaland/ western Indochina clade of *Leptobrachium* as a member of *L. smithi* species complex. It is grouped with high node support (1.0/100, hereafter node support values are given for BI PP/ BS ML, respectively) with the "*Leptobrachium* sp. 4" sample of *Matsui et al. (2010)* from Pilok District of Kanchanaburi Province, and together they unambiguously form a sister mtDNA lineage to the group including *L. smithi* and *L. rakhinense* (1.0/100, see Fig. 2).

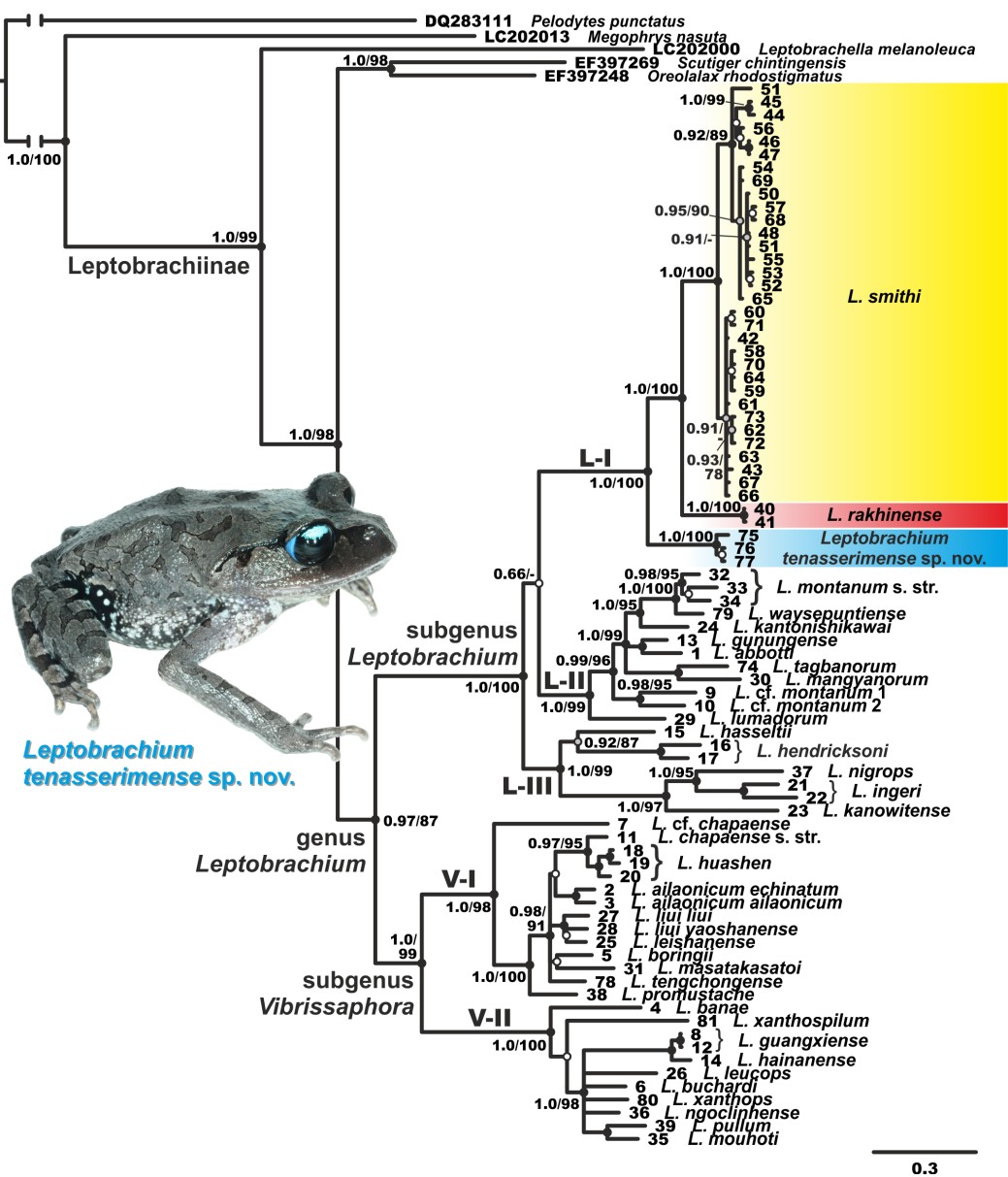

**Figure 2 Phylogenetic BI tree of *Leptobrachium* reconstructed on the base of 2,494 bp (partial *12S rRNA- tRNA*val*-16S rRNA* sequences).** Values on the branches correspond to BI PP/ML BS, respectively; black, grey and white circles correspond to well-supported, moderately supported and non-supported nodes, respectively. Color marking of species in *L. smithi* species group corresponds to Fig. 1. For specimen and locality information see Table 1. Photo by Nikolay A. Poyarkov.

## Genetic divergence

The uncorrected pairwise divergences in the 16S rRNA gene fragment within and among *Leptobrachium* species examined in our analysis are summarized in Table S2. The observed interspecific genetic distances ranged from $p = 0.9\%$ (between *L. hainanense* Ye & Fei and *L. guangxiense* Fei, Mo, Ye & Jiang) to 23.6% (between *L. promustache* (Rao, Wilkinson &

Zhang) and *Leptobrachium* sp. from Tenasserim). The undescribed *Leptobrachium* species from Tenasserim was found to be most closely related to *L. smithi* ($p = 10.4\%$) and *L. rakhinense* ($p = 10.5\%$). These levels of divergences in 16S rRNA gene are notably higher than those observed between many recognized sister species of *Leptobrachium* (0.9%–3.2%), as well as the highest intraspecific genetic distance (4.6%, see Table S2). Intraspecific genetic variation in the 16S rRNA gene fragment was 0.0–0.8% in the undescribed *Leptobrachium* species from Tenasserim, 0.0% in *L. rakhinense*, and 0.0–1.3% in *L. smithi*, respectively.

## Systematics

Our analysis unambiguously suggests that the population of *Leptobrachium* sp. from the Tenasserim Region mountains in western Thailand is a member of the *L. smithi* species complex and represents a distinct highly-divergent mtDNA lineage with sister relationship to the group including *L. smithi* and *L. rakhinense*. As we show below, the observed molecular differences correspond to the differences in external morphological traits, which are considered to be useful for the diagnostics of *Leptobrachium* species, and allow to distinguish *Leptobrachium* sp. from all other congeners. Based on molecular and morphological lines of evidence, we hence consider the *Leptobrachium* sp. from the Tenasserim as a new species and describe it herein.

### *Leptobrachium tenasserimense* sp. nov.
(Figs. 3–5; Table 1 and Table S2)

**Chresonymy**
"*Leptobrachium* sp. 4"—*Matsui et al., 2010*: 263.

**Holotype.** AUP-00362, an adult male, collected from montane forest of Khao Laem Mt., Suan Phung District, Ratchaburi Province, western Thailand (13°32′50.35″N, 99°12′14.18″E; elevation 715 m a.s.l.) on September 8, 2017, by Parinya Pawangkhanant (Fig. 3).

**Paratypes.** Five adult males: AUP-00360, AUP-00361, AUP-01284 (field ID NAP-06598), AUP-01285 (field ID NAP-06599) and ZMMU A-5919 (field ID NAP-06600); and two adult females: AUP-01283 (field ID NAP-06596) and ZMMU A-5918 (field ID NAP-06597); all specimens with collection information same as for the holotype.

**Etymology.** The specific name is a Latinized toponymic adjective in neutral gender derived from "Tenasserim"—a historical name of the region in the northern part of the Malayan Peninsula in southern Indochina, and for the mountain chain known as "Tenasserim Hills", where the new species occurs.

**Diagnosis.** A member of the genus *Leptobrachium* on the basis of head width being larger than tibia length; skin dorsally with a network of ridges; oval and large axillary glands present; extremities of digits rounded; breeding males lacking spines on fingers and breast; and bicolored iris (*Yang, Wang & Chan, 2016*). The new species can be distinguished from other congeners by the following combination of morphological characteristics: (1) medium-sized species, with adult SVL of 41.4–58.8 mm in males and 54.7–58.6 mm in

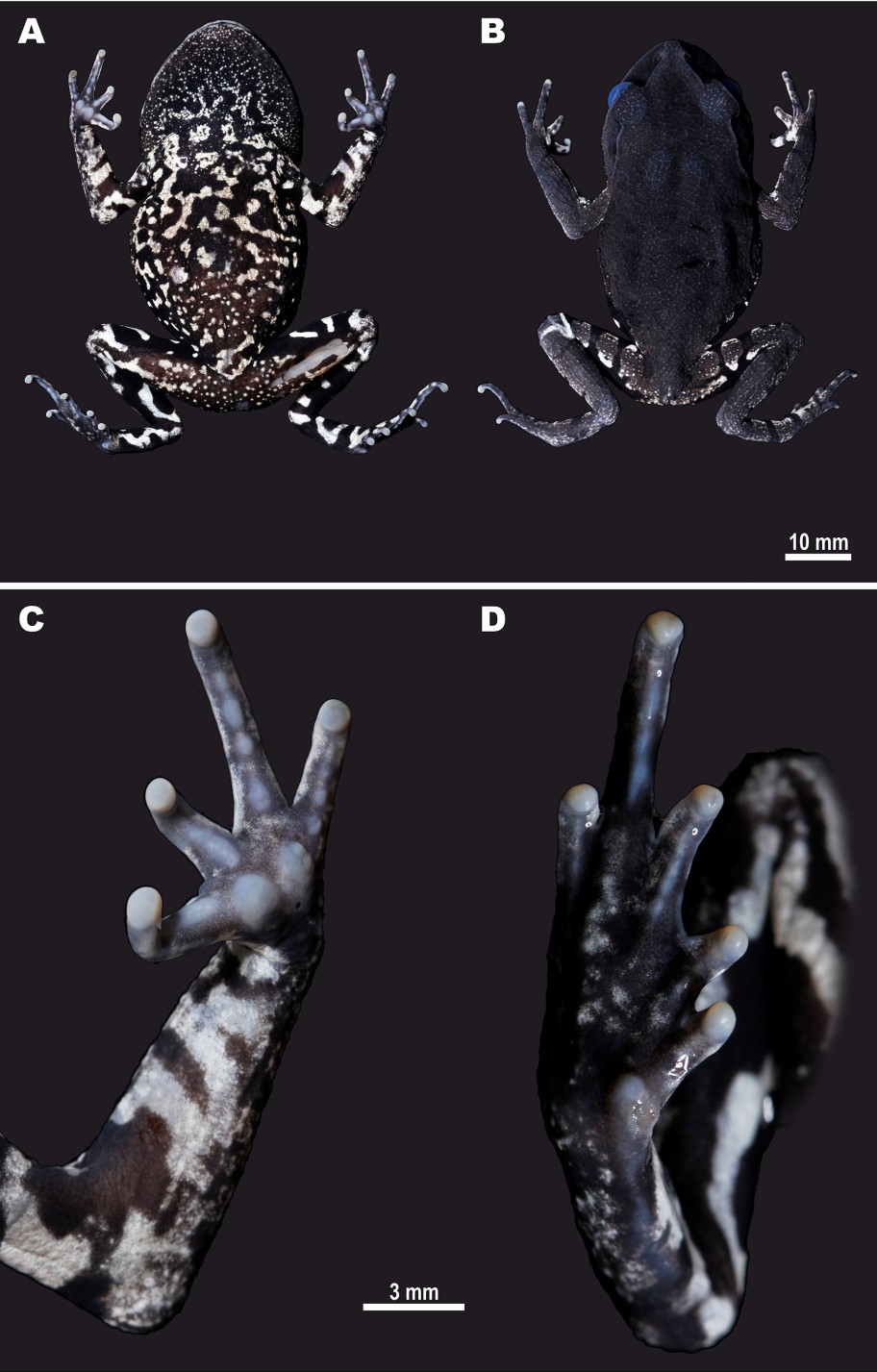

**Figure 3** **Male holotype of *Leptobrachium tenasserimense* sp. nov. (AUP-00362) after preservation.**
(A) Ventral view; (B) dorsal view; (C) volar view of left hand; (D) palmar view of right foot. Photos by
Parinya Pawangkhanant.

females; (2) rounded finger and toe tips; (3) relative finger lengths: II<IV<I<III; relative toe lengths: I<II<V<III<IV; (4) toe webbing thick and well developed; (5) inner metatarsal tubercle comparatively small; (6) iris bicolored, black ventrally and turquoise dorsally, with light blue sclera; (7) dorsum brown to grey with distinct darker markings edged with dark-brown, dark head markings usually distinct; (8) belly and limbs ventrally whitish with dense contrasting confluent black blotches and reticulations; (9) tympanum free of dark marking or dark coloration covering only the uppermost one-third of tympanum; (10) dark canthal stripe present, narrow, not covering loreal region; (11) ventro-lateral row of dark spots or blotches absent; (12) limbs, including fingers and toes, dorsally with distinct dark bars; tibia with four to five dark transverse bars; (13) dense dark reticulations or large dark blotches at groin continuing to ventral and posterior sides of thighs; (14) femoral gland in shape of large white rounded blotch; (15) males with single vocal sac, mature males lack lip spinules.

**Description of holotype.** Adult male with SVL 58.8 mm; habitus robust, body slightly tapering to groin (Figs. 3A, 3B). **Head.** Head broad and flattened, slightly wider (HW to SVL ratio 41.7%) than long (HL to SVL ratio 40.5%), HW/HL ratio 102.9%; snout gently rounded in dorsal view, sharply sloping in profile, notably projecting beyond lower jaw in lateral view; nostrils round with a dorsolateral orientation, located below canthus rostralis, closer to tip of snout than to eye (N-EL to SVL ratio 10.6%); canthus rostralis distinct, sharp; loreal region slightly concave, steep; internarial distance (IND to SVL ratio 7.6%) twice shorter than interorbital distance (IOD to SVL ratio 16.1%), IND/IOD ratio 47.1%; eyes large, bulging, distinctly projecting from sides of head in lateral and dorsal views, eye diameter slightly smaller (EL to SVL ratio 12.7%) than snout length (SL to SVL ratio 18.2%); interorbital distance (IOD to SVL ratio 16.1%) two times greater than upper eyelid width (UEW to SVL ratio 8.1%); pineal ocellus absent; tympanum distinct, round, not depressed relative to surrounding area, tympanum diameter (TD to SVL ratio 5.3%) almost a half of eye diameter (TD/EL ratio 41.6%), comprising ca. two thirds of the distance between tympanum and eye (TD/TEL ratio 67.5%); no vomerine teeth; tongue large, broad and unnotched; vocal sac single, gular. **Limbs.** Forelimbs slender, long; fingers slender (Fig. 3C), free of webbing; finger tips rounded and slightly swollen; relative finger lengths: II<IV<I<III; all fingers with weak dermal fringes to tips; two strong semi-circular palmar tubercles, prominent, bulging, contacting each other medially, much larger than finger tips, inner palmar tubercle slightly smaller than outer palmar tubercle; callous tissue forming low ridges on ventral surfaces of all fingers, subdigital ridges distinctly divided into three subarticular tubercles on finger III and into two subarticular tubercles on finger IV; nuptial pads absent. Hindlimbs relatively short, slender (HLL to SVL ratio 114.7%); heels not contacting when legs held at right angles to body; tibia distinctly longer (TL to SVL ratio 35.2%) than foot (FL to SVL ratio 27.0%), FL/TL ratio 76.6%; tibiotarsal articulation of adpressed limb reaching to the level of middle of tympanum. Toes slightly flattened (Fig. 3D), all with thick dermal lateral fringes reaching to the toe tips; toe webbing thick and well-developed, toe webbing formula: I 1–2 II 1–3 III 1–3$\frac{1}{2}$ IV 3$\frac{3}{4}$ –1$\frac{1}{2}$ V. Tips rounded, slightly swollen; relative toe lengths: I<II<V<III<IV. Subarticular tubercles indistinct, replaced by elongated low callous ridges; inner metatarsal tubercle distinct, oval,

comparatively small (IML to SVL ratio 4.9%) comprising ca. 30.0% of distance between tip of toe I and tubercle; outer metatarsal tubercle absent. **Skin.** Cornified spinules or spines on upper lip absent. Skin dorsally with distinct network of dermal ridges, with minute granules scattered on dorsal surface of head, especially on interorbital region, granules getting denser on upper eyelid; ventral surfaces weakly granular, granules getting more distinct on belly and body flanks; supratympanic ridge low, distinct, running from posterior corner of eye to axilla; limbs smooth ventrally, with dermal ridges dorsally, forming dense longitudinal rows on upper arm; flat axillary gland barely distinct at medial border of axilla behind arm insertion; femoral gland flat, small, rounded, present on distal half of posteroventral surface of thigh closer to knee than to vent.

**Measurements of holotype (all in mm).** SVL 58.8; HL 23.8; SNL 4.9; NEL 6.2; SL 10.7; EL 7.5; TEL 4.6; TD 3.1; HW 24.5; IND 4.5; ICD 9.8; IOD 9.4; UEW 4.8; UEMD 20.4; FFL 6.9; OPTL 2.4; IPTL 1.5; HAL 11.2; FAW 4.1; HLL 67.4; TL 20.7; FL 15.8; FTL 9.2; IMTL 2.9.

**Coloration of holotype in life.** In life, dorsally grayish-brown with three large dark-brown blotches in interorbital region forming an inverted V-shaped pattern and less distinct dark blotches on dorsum; all blotches edged with dark-brown; a fine black canthal stripe running along canthus rostralis from anterior corner of eye to nostril bifurcating ventrally posterior to nostril; loreal region and upper jaw brown; snout tip light-gray; thin dark-brown line running from each nostril towards tip of snout; a prominent narrow and solid black supratympanic stripe running along the supratympanic ridge from posterior corner of eye to mouth angle, getting wider above tympanum and covering the dorsal one-fourth of tympanum; dorsal background grayish in color laterally fading to lighter whitish color towards belly; axilla and groin with numerous irregular dark blotches; ventral surfaces with white to bluish-white background color heavily reticulated with contrasting black confluent blotches, getting denser posteriorly and towards groin (Fig. 3D); limbs dorsally dark-gray with distinct black bars: three bars on each thigh, four to five less distinct bars on shanks, two to three dark bars on tibiotarsus; ventrally limbs with numerous irregular black and white blotches; posterior surfaces of thighs banded with black bars confluent with dark coloration on groin and flanks; small white spots scattered around vent. Iris bicolored, black to dark brown in lower two-thirds, whitish-turquoise in upper one-third; scleral arc bright light blue.

**Coloration of holotype in preservative.** In preservative (Fig. 3) the general coloration pattern did not change significantly, but dorsal brown coloration faded to dark-grey, bluish or cream tints and ventral coloration faded to white; black blotches on ventral surface, groin and posterior surfaces of thighs are distinct. Iris and sclera coloration completely faded.

**Variation.** Morphological variation of the type series is presented in Table 1. In general, morphology and coloration of the type species are similar with that described for the holotype. Males have smaller body size (mean SVL 49.3 ± 4.8; $N = 6$) than female paratypes (mean SVL 56.7 ± 2.0; $N = 2$); SDI index equals 1.15. Details of paratype female (ZMMU A-5918) coloration in life are shown in Fig. 4. In life, females have much paler throat coloration and less contrasting black and white blotched pattern on belly (Fig. 4A);

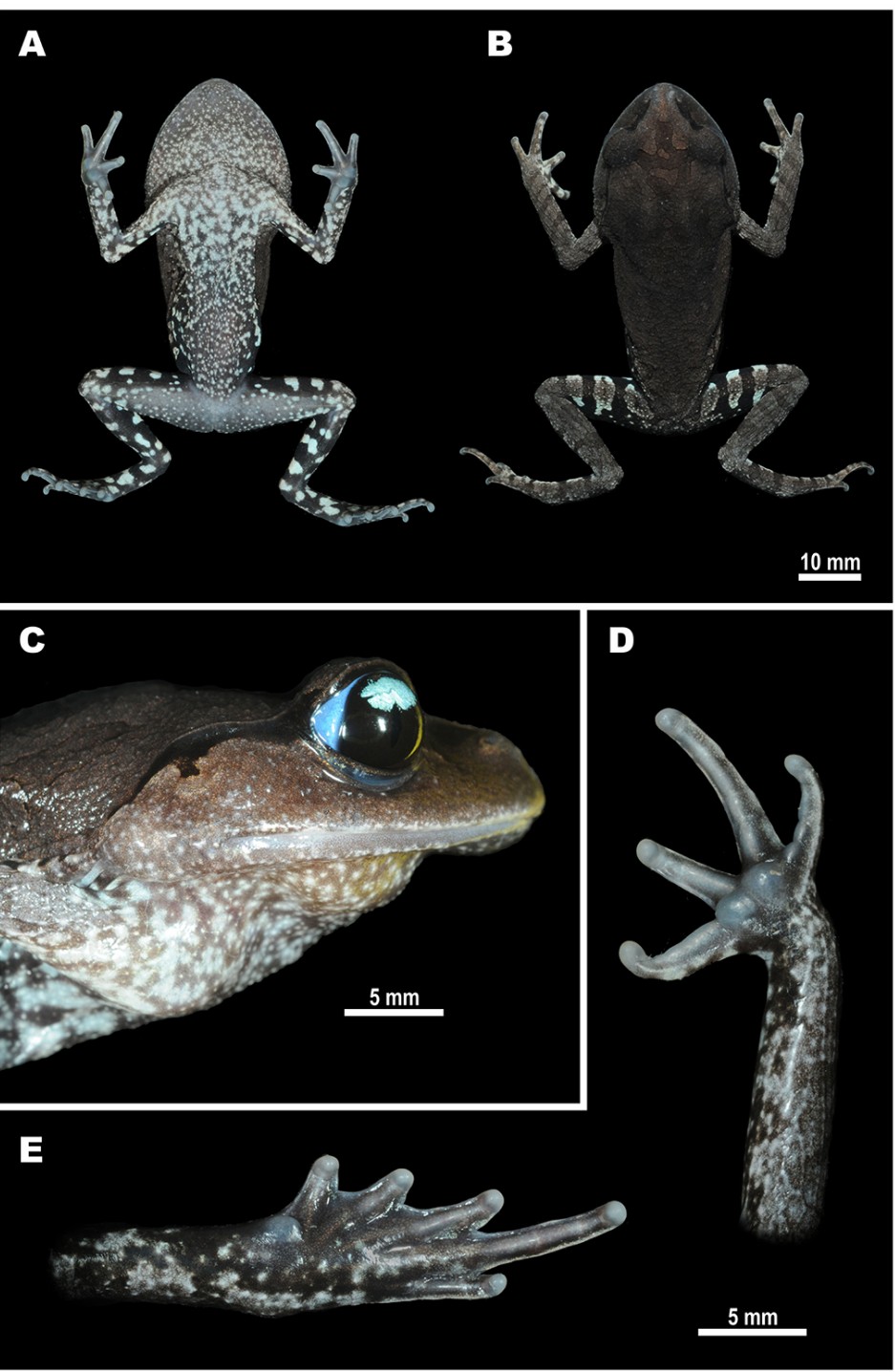

**Figure 4** **Female paratype of *Leptobrachium tenasserimense* sp. nov. (ZMMU A-5918) in life.** (A) Ventral view; (B) dorsal view; (C) lateral view of head; (D) volar view of left hand; (E) palmar view of left foot. Photos by Nikolay A. Poyarkov.

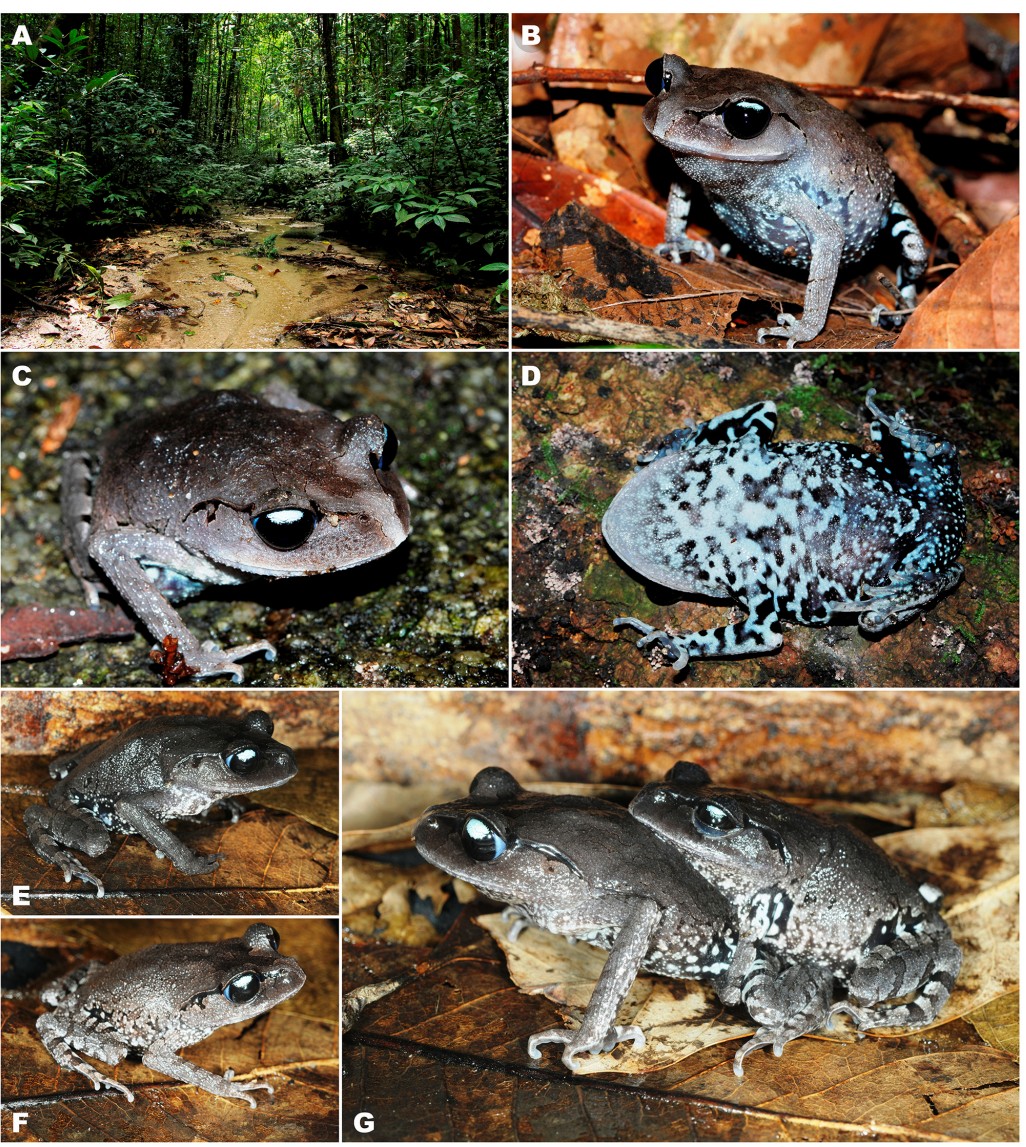

**Figure 5  Color variation of *Leptobrachium tenasserimense* sp. nov. in life.** (A) Natural habitat at the type locality in Khao Laem Mountain, Suan Phung District, Ratchaburi Province; (B) and (C) dorsolateral views of adult male (not collected) *in situ*; (D) ventral view of adult male (not collected) *in situ*; (E) male paratype ZMMU A-5919; (F) female paratype ZMMU A-5918; (G) amplexus *in situ*. Photos (A–D) by Parinya Pawangkhanant; (E–G) by Nikolay A. Poyarkov.

females also tend to have slightly lighter coloration of dorsum than males (Figs. 4B, 5). The size of dark blotch in tympanal region varies from comparatively large black blotch confluent with dark supratympanic stripe covering the dorsal one third of tympanum (as in paratype male ZMMU A-5919), to smaller dark stripe touching the dorsal edge and adjacent one fourth of tympanum (as in Figs. 5B, 5C) to complete absence of dark markings on tympanum (as in paratype female ZMMU A-5918; see Figs. 5F, 5G).

**Distribution.** Currently known only from two localities in the northern part of Tenasserim: from the type locality in Suan Phung District, Ratchaburi Province (this work), and from Pilok District in Kanchanaburi Province (*Matsui et al., 2010*) (see Fig. 1). Occurrence in Phetchaburi Province of Thailand and in the adjacent parts of Tanintharyi Division of Myanmar is strongly anticipated.

**Ecology and Natural history.** Specimens of the new species were recorded along a slow-flowing stream in a montane tropical forest on Khao Laem Mountain at elevations from 700 to 1000 m a.s.l. (see Fig. 5A). The multi-species codominant (polydominant) tropical forest at the type locality had dense vegetation with tangles of the giant bamboo (*Dendrocalamus asper* (Schult.) Backer). Frogs were observed in leaf litter or under tree roots; males were calling during our field observations in August, September and November 2017. Amplexus was recorded in November 2017 (see Fig. 5G).

Herpetofauna species recorded sympatrically with the new species at the type locality include: *Leptobrachium smithi*, *Xenophrys* cf. *major* (Boulenger), *Leptobrachella melanoleuca* (Matsui), *Leptobrachella fuliginosa* (Matsui), *Amolops panhai* Matsui & Nabhitabhata, *Alcalus tasanae* (Smith), *Limnonectes jarujini* Matsui, Panha, Khonsue & Kuraishi, *Limnonectes doriae* (Boulenger), *Limnonectes macrognathus* (Boulenger), *Microhyla berdmorei* (Blyth), *Acanthosaura crucigera* Boulenger, *Pseudoxenodon macrops* (Blyth), *Trimeresurus popeiorum* Smith, and *Rhabdophis chrysargos* (Schlegel). At the type locality of the new species in Khao Laem Mountain *L. smithi* was recorded in the same biotopes as *Leptobrachium tenasserimense* sp. nov. at elevations around 800 to 1,200 m a.s.l. and the two species shared same streams for reproduction and the breeding season of two species seem to overlap. Additional studies are required to elucidate reproductive biology and ecology of two sympatric *Leptobrachium* species of Khao Laem Mountain.

**Comparisons.** Together with *L. smithi* and *L. rakhinense*, the new species belongs to subclade L1 of the Sundaland/Thailand clade (subgenus *Leptobrachium*), which occurs in southern Myanmar, Thailand, Peninsular Malaysia, Sumatra, Java, Bali, Borneo, and the Philippines. Unrelated island *Leptobrachium* taxa are omitted from comparisons for simplicity. Thus, we compared *Leptobrachium tenasserimense* sp. nov. to the mainland members of the subgenus *Leptobrachium*, and to all other recognized species of *Leptobrachium* from Thailand and surrounding parts of Indochina which belong to the subgenus *Vibrissophora*.

Comparisons of the new species with the members of subclade L1 of the Sundaland/Thailand clade (subgenus *Leptobrachium*), namely with *L. smithi* and *L. rakhinense*, appear to be the most pertinent. *Leptobrachium tenasserimense* sp. nov. can be distinguished from *L. smithi* (from Thailand, western Laos, northern peninsular Malaysia and easternmost Myanmar) by the following combination of morphological attributes: black and turquoise bicolored iris (vs. black and red or black and bright yellow bicolored iris in *L. smithi*), narrow dark canthal stripe (vs. broad dark canthal stripe covering narial area in *L. smithi*), absence of dark markings in tympanal area (vs. tympanum covered with dark markings in *L. smithi*), presence of distinct dark markings on head and dorsum (vs. dorsum dark gray with no distinct markings in *L. smithi*), contrasting black and white ventral coloration that uniformly covers throat, chest, belly and ventral surfaces of limbs (vs. mostly whitish

Pawangkhanant et al. (2018), *PeerJ*, DOI 10.7717/peerj.5584

**Table 1** **Selected measurements (in mm) of *Leptobrachium tenasserimense* sp. nov. type series.** For character abbreviations see 'Materials and Methods'.

| Specimen | Type status | Sex | SVL | HL | S-NL | N-EL | SL | EL | T-EL | TD | HW | IND | ICD | IOD | UEW | UEMD | FFL | OPTL | IPTL | HAL | FAW | HLL | TL | FL | FTL | IMTL |
|---|---|---|---|---|---|---|---|---|---|---|---|---|---|---|---|---|---|---|---|---|---|---|---|---|---|---|
| AUP-00362 | (Holotype) | m | 58.8 | 23.8 | 4.9 | 6.2 | 10.7 | 7.5 | 4.6 | 3.1 | 24.5 | 4.5 | 9.8 | 9.4 | 4.8 | 20.4 | 6.9 | 2.4 | 1.5 | 11.2 | 4.1 | 67.4 | 20.7 | 15.8 | 9.2 | 2.9 |
| AUP-00361 | (Paratype) | m | 52.5 | 24.2 | 5.1 | 6.6 | 10.1 | 6.3 | 9.0 | 2.9 | 26.7 | 5.0 | 9.6 | 7.4 | 4.9 | 23.6 | 6.4 | 2.0 | 2.0 | 11.0 | 3.0 | 64.1 | 15.4 | 18.7 | 17.1 | 2.1 |
| AUP-00360 | (Paratype) | m | 51.3 | 23.8 | 4.1 | 5.2 | 9.9 | 6.5 | 3.5 | 2.8 | 21.2 | 5.0 | 9.8 | 9.4 | 4.3 | 13.2 | 4.9 | 2.2 | 1.9 | 10.8 | 3.7 | 43.4 | 19.7 | 16.7 | 11.1 | 2.6 |
| AUP-01285 | (Paratype) | m | 41.4 | 15.8 | 3.7 | 4.0 | 6.7 | 6.1 | 3.7 | 2.1 | 17.6 | 3.6 | 7.1 | 6.5 | 4.1 | 13.2 | 4.1 | 1.5 | 1.1 | 7.7 | 2.2 | 47.9 | 15.9 | 13.1 | 8.4 | 1.6 |
| AUP-01284 | (Paratype) | m | 45.2 | 17.2 | 4.1 | 4.1 | 7.4 | 6.1 | 2.9 | 2.7 | 19.4 | 3.9 | 7.8 | 6.2 | 4.3 | 14.6 | 5.3 | 1.9 | 1.7 | 9.1 | 2.4 | 50.2 | 17.2 | 13.7 | 9.0 | 1.6 |
| ZMMU A-5919 | (Paratype) | m | 46.9 | 19.5 | 3.3 | 4.3 | 6.9 | 6.6 | 3.0 | 3.3 | 20.0 | 4.0 | 9.3 | 7.0 | 4.5 | 16.0 | 5.2 | 1.9 | 1.3 | 10.9 | 3.5 | 56.6 | 17.4 | 15.8 | 7.5 | 1.5 |
| **Males (N = 6)** | | **mean** | **49.3** | **20.7** | **4.2** | **5.1** | **8.6** | **6.5** | **4.4** | **2.8** | **21.6** | **4.3** | **8.9** | **7.6** | **4.5** | **16.8** | **5.5** | **2.0** | **1.6** | **10.1** | **3.1** | **54.9** | **17.7** | **15.6** | **10.4** | **2.0** |
| | | *SD* | *4.8* | *3.2* | *0.6* | *0.9* | *1.6* | *0.3* | *1.6* | *0.3* | *2.7* | *0.5* | *1.0* | *1.2* | *0.2* | *3.4* | *0.8* | *0.2* | *0.3* | *1.1* | *0.6* | *7.8* | *1.7* | *1.5* | *2.5* | *0.5* |
| AUP-01283 | (Paratype) | f | 54.7 | 24.1 | 4.2 | 5.7 | 8.9 | 7.1 | 4.6 | 3.0 | 22.5 | 4.4 | 8.9 | 9.0 | 4.7 | 16.6 | 5.8 | 1.7 | 1.3 | 10.4 | 3.0 | 60.0 | 20.0 | 15.2 | 10.2 | 1.8 |
| ZMMU A-5918 | (Paratype) | f | 58.6 | 25.3 | 4.2 | 4.8 | 8.7 | 7.7 | 3.2 | 4.5 | 25.6 | 4.7 | 11.7 | 8.5 | 5.6 | 18.5 | 5.7 | 2.1 | 1.6 | 12.6 | 3.7 | 72.8 | 21.2 | 18.4 | 10.3 | 1.9 |
| **Females (N = 2)** | | **mean** | **56.7** | **24.7** | **4.2** | **5.3** | **8.8** | **7.4** | **3.9** | **3.8** | **24.1** | **4.6** | **10.3** | **8.8** | **5.2** | **17.6** | **5.8** | **1.9** | **1.5** | **11.5** | **3.4** | **66.4** | **20.6** | **16.8** | **10.2** | **1.8** |
| | | *SD* | *2.0* | *0.6* | *0.0* | *0.5* | *0.1* | *0.3* | *0.7* | *0.8* | *1.5* | *0.1* | *1.4* | *0.3* | *0.4* | *0.9* | *0.0* | *0.2* | *0.1* | *1.1* | *0.4* | *6.4* | *0.6* | *1.6* | *0.0* | *0.0* |

**Notes.**

SD, standard deviation

belly background coloration with dark speckles occurring posteriorly in *L. smithi*), finger II shorter than finger IV (vs. finger IV shorter than finger II in *L. smithi*), and comparatively smaller SDI = 1.15 (vs. SDI = 1.34 in *L. smithi*). Furthermore, *L. tenasserimense* sp. nov. can be distinguished by posterior surfaces of thighs having white spots (vs. uniformly black posterior surfaces of thighs in *L. smithi*).

*Leptobrachium tenasserimense* sp. nov. is morphologically similar to *L. rakhinense* from Rakhine State in Myanmar but can be distinguished by the following combination of morphological attributes: black and turquoise bicolored iris with light blue sclera (vs. black and red bicolored iris with light blue sclera in *L. rakhinense*), narrow dark canthal stripe (vs. broad dark canthal stripe broadly covering narial area in *L. rakhinense*), absence of dark markings in tympanal area (vs. tympanum covered with dark coloration in *L. rakhinense*), gray dorsum with darker blotches outlined with dark-brown (vs. light gray to brown dorsum with distinct dark brownish blotches outlined with white in *L. rakhinense*), ventral coloration with white background covered with contrasting confluent black blotches (vs. light venter with white speckles in *L. rakhinense*). Furthermore, *Leptobrachium tenasserimense* sp. nov. can be distinguished by absence of lateral series of dark spots (vs. distinct in *L. rakhinense*); also, new species possesses very distinctive fore- and hind leg stripes, while in *L. rakhinense* leg bars are not as distinctive.

By the absence of spines on the upper lip in sexually active males, *Leptobrachium tenasserimense* sp. nov. distinctly differs from those congeners which were formerly referred to as the genus *Vibrissaphora*, i.e., *L. ailaonicum* (Yang, Chen & Ma) (northern Vietnam and Yunnan), *L. promustache* (northern Vietnam and Yunnan) and *L. ngoclinhense* (Orlov) (central Vietnam), all of which are reported to possess cornified spines on the upper lip in sexually active males (*Rao, Wilkinson & Zhang, 2006*; *Fei & Ye, 2005*; *Fei et al., 2009*; *Yang, Wang & Chan, 2016*).

Eye coloration in life is reported to be an important diagnostic characteristic for species identification of the genus *Leptobrachium* (*Hamidy & Matsui, 2010*; *Matsui, Nabhitabhata & Panha, 1999*; *Stuart et al., 2011*; *Stuart et al., 2012*; *Wogan, 2012*; *Yang, Wang & Chan, 2016*). By having a bicolored iris with lower one-third black and upper one-third turquoise with light-blue sclera, *Leptobrachium tenasserimense* sp. nov. can be distinguished from the following members of the subgenus *Vibrissaphora*: *L. chapaense* (northern Vietnam and Yunnan; iris uniformly dark brown with black sclera), *L. huashen* (Yunnan and northern Thailand; iris bicolored with white upper one-third or uniformly dark brown with black sclera, but see discussion in *Yang, Wang & Chan, 2016*, *L. guangxiense* (Guangxi Province of China, central and northern Vietnam; iris bicolored with upper one-fourth to one-third white or bluish-white with black sclera, see *Chen et al., 2013*), *L. masatakasatoi* Matsui (northern Laos and adjacent Vietnam; iris uniform brown or bicolored with white upper one-fourth; sclera black; see *Pham et al., 2016*); *L. xanthops* Stuart, Phimmachak, Seateun & Sivongxay (Bolaven Plateau, Laos; upper half of iris pale yellow with whitish sclera), *L. buchardi* Ohler, Teynié & David (Bolaven Plateau, Laos; upper third of iris whitish-green to turquoise with bright blue sclera), *L. leucops* Stuart, Rowley, Tran, Le & Hoang (Langbian Plateau, Vietnam; upper part of iris white with gray or dark sclera), *L. xanthospilum* Lathrop, Murphy, Orlov & Ho (central Vietnam; upper third of iris white,

sclera dark-brown); *L. banae* Lathrop, Murphy, Orlov & Ho (central Vietnam; upper third of iris whitish, sclera white); *L. bompu* Sondhi & Ohler (Himalaya; iris uniformly grey-blue, dark sclera), *L. mouhoti* Stuart, Sok & Neang and *L. pullum* (Smith) (southern and central Vietnam; iris black or dark-brown with an orange-yellow or red sclera), *L. ngoclinhense* (central Vietnam; iris uniformly dark brown). By having gray dorsum with darker blotches outlined with dark-brown and having whitish venter with contrasting confluent black blotches the new species can be further distinguished from *L. banae* (reddish dorsum, reddish bands on limbs), *L. xanthospilum* (large, yellow, glandular spots on the flanks), *L. mouhoti* and *L. pullum* (dark uniform brownish or blackish coloration, belly light-gray with dark mottling posteriorly), *L. xanthops, L. leucops, L. masatakasatoi, L. chapaense, L. guangxiense, L. huashen* (all have dark belly with no contrasting white and black pattern) and *L. buchardi* (light greyish belly without dark patterning).

From the mainland members of the subgenus *Leptobrachium* the new species can be distinguished by the following combination of morphological attributes. From *L. nigrops* (Singapore, southern peninsular Malaysia, Sumatra and an unconfirmed record from Thailand) the new species can be easily distinguished by having a bicolored iris (vs. uniform black or dark brown iris in *L. nigrops*), larger adult body size with male SVL 41.4–58.8 mm, female SVL 54.7–58.6 mm (vs. male SVL 24.9–40.1 mm, female 33.7–42.7 mm in *L. nigrops*) and gray dorsum with darker blotches outlined with dark-brown (vs. light-gray to beige dorsum with large brown blotches and longitudinal stripes in *L. nigrops*).

*Leptobrachium tenasserimense* sp. nov. can be distinguished from *L. hendricksoni* by having a black and turquoise bicolored iris (vs. iris uniformly pale red or black and red bicolored iris in *L. hendricksoni*), a narrow dark canthal stripe (vs. broad dark canthal stripe covering narial area in *L. hendricksoni*), distinct dark markings on head and dorsum (vs. head and dorsal markings absent or indefinite in *L. hendricksoni*), by lack of dark markings in tympanal area (vs. tympanum covered with dark markings in *L. hendricksoni*), by having four to five dark tibial bars (vs. no tibial bars in *L. hendricksoni*), by having contrasting whitish belly with dark confluent pattern (vs. creamy belly with black speckles in *L. hendricksoni*) and by females being slightly larger than males, SDI 1.15 (vs. females much larger than males, SDI 1.38 in *L. hendricksoni*).

## DISCUSSION

Our study agrees with previous views on mtDNA genealogic relationships within the genus *Leptobrachium* (see *Matsui et al., 2010*; *Yang, Wang & Chan, 2016*), suggesting its subdivision into two major geographic groups, corresponding to (1) Sundaland—western Indochina and (2) southern China—eastern Indochina centers of diversification. The *L. smithi* species complex is a monophyletic group within the first clade inhabiting western Indochina, southern Myanmar and northern part of Malayan Peninsula (see Fig. 1). Our work further indicates the presence of yet undescribed diversity within the genus *Leptobrachium*: in addition to the new species described herein, two lineages from Borneo tentatively indicated as *L.* cf. *montanum* 1 and 2 (samples 9 and 10; see Fig. 2, Table S1), and one lineage from northern Vietnam tentatively indicated as *L.* cf. *chapaense* (sample

11; see Fig. 2, Table S1), likely correspond to yet undescribed species. Further studies are required to clarify their taxonomic status.

Iris and sclera coloration in life is recognized as a useful diagnostic character for species identification in the genus *Leptobrachium* (*Matsui, Nabhitabhata & Panha, 1999*; *Matsui et al., 2010*; *Hamidy & Matsui, 2010*; *Sondhi & Ohler, 2011*; *Stuart et al., 2011*; *Stuart et al., 2012*; *Wogan, 2012*; *Yang, Wang & Chan, 2016*). *Matsui et al. (2010)* analyzed possible evolution of the eye coloration in *Leptobrachium* and suggested that bicolored white and black iris was characteristic for the common ancestor of the China–East Indochina Clade (subgenus *Vibrissaphora*). The ancestral iris coloration for the Sundaland–West Indochina Clade (subgenus *Leptobrachium* sensu stricto) was reconstructed as uniform black, and black and white bicolored iris was not recorded in this clade up to date (*Matsui et al., 2010*). All members of the *L. smithi* complex share a bluish (to some extent) coloration of scleral arc, but differ in iris coloration: *L. smithi* has a black and orange or black and yellow bicolored iris; *L. rakhinense* has a black and red bicolored iris, while *Leptobrachium tenasserimense* sp. nov. has a black and almost white (light-turquoise) bicolored iris. This suggests that the black and white bicolored iris coloration is also present in the Sundaland—West Indochina Clade of *Leptobrachium* and likely corresponds to the ancestral condition for the genus.

Distribution of members of the *L. smithi* complex is shown in Fig. 1. *Leptobrachium rakhinense* is restricted to the Rakhine Yoma Mountain Range in south-western Myanmar (*Wogan, 2012*). Distribution of this species extends northwards to Mizoram and Meghalaya states of India and Bangladesh (*Chanda, 2002*; *Ahmed, Das & Dutta, 2009*; *Mahony et al., 2009*); however, the taxonomic status of populations outside Myanmar is tentative and requires clarification. *L. smithi* has a wide distribution in western Indochina extending from western Laos to northern Thailand, and southwards along the Tenasserim Mountain Range across the Isthmus of Kra to southernmost Thailand and Malaysian Island of Langkawi (*Matsui, Nabhitabhata & Panha, 1999*; *Matsui et al., 2010*; *Grismer et al., 2006*); this species was also recorded in the adjacent parts of Myanmar, including Shan and Mon states and Tanintharyi Division (*Wogan, 2012*). *Leptobrachium tenasserimense* sp. nov. occurs only in montane areas of northern Tenasserim and is currently recorded from two provinces in western Thailand: Ratchaburi and Kanchanaburi. The new species is also likely to inhabit Phetchaburi Province in northern Tenasserim, as well as adjacent parts of Tanintharyi Division of Myanmar. Further field surveys are required for estimation of *Leptobrachium tenasserimense* sp. nov. distribution extent, which is crucial for the determination of the IUCN conservation status of the new species.

It is remarkable that the new species is sympatrically distributed with *L. smithi* both in Ratchaburi and Kanchanaburi provinces. Both species were recorded sharing the same biotopes in Ratchaburi Province, though *Leptobrachium tenasserimense* sp. nov. prefers higher elevations and less disturbed montane forests and never occurs lower than 700 m a.s.l., while *L. smithi* is recorded from a much wider range of elevations (ca. 10–1,500 m a.s.l.). The ecology and distribution of these two species in the Tenasserim Region requires additional investigation. Since *Leptobrachium tenasserimense* sp. nov. is suggested to be a sister lineage to other members of the *L. smithi* species complex, it is likely that the

northern Tenasserim Region played an important role in the diversification of this lineage of *Leptobrachium*. It is noteworthy that a similar biogeographic pattern was also recently reported for the genus *Leptobrachella* (see *Chen et al., 2018*).

## CONCLUSIONS

Our new discovery of *Leptobrachium tenasserimense* sp. nov. indicates that the montane forests of northern Tenasserim Region on the border of Thailand and Myanmar contain herpetofaunal diversity that is still unrecognized. This comparatively narrow area is known for an exceptionally high number of endemic species of amphibians and reptiles discovered by recent herpetofaunal surveys (*Mulcahy et al., 2018*), including a new genus and species of microhylid frogs (*Suwannapoom et al., 2018*), two new species of megophryid frogs (*Matsui, 2006*), two new species of bufonid frogs (*Wilkinson, Sellas & Vindum, 2012*; *Matsui, Khonsue & Panha, 2018*), five endemic gecko species and two endemic species of snakes (see *Sumontha et al., 2017*). Possible reasons behind such exceptional herpetofaunal endemism are yet unclear; recent studies indicate that the northern part of Tenasserim Region played a key role in the faunal exchange between Sundaland and the mainland Indochina during the Cenozoic (see *Chen et al., 2018* for discussion). Our study provides further evidence for the hidden biodiversity of the Tenasserim Region, and suggests that its herpetofauna is still clearly underestimated. Further field surveys are required for facilitating herpetological exploration and elaboration of measured conservation of this hidden diversity.

## ACKNOWLEDGEMENTS

We would like to thank the Laboratory Animal Research Center, University of Phayao and The Institute of Animal for Scientific Purposes Development (IAD), Thailand for the permission to work in the field. We want to thank the Rabbit in the Moon foundation for help during the field work; we thank Kanokwan Yimyoo for constant assistance in the field and in the lab and Pattarawhich Dawwrueng for assistance, and Thiti Ruengsuwan, Kawin Jiaranaisakul, Akkrachai Aksornneam for help during the field work. NAP thanks Valentina F. Orlova and Roman A. Nazarov (Zoological Museum of Moscow State University) for help during the work in collection under their care, Evgeniya N. Solovyeva (Zoological Museum of Moscow University) for help with data analyses, Evgeniy S. Popov (Botanical Institute R.A.S., St. Petersburg) for providing map and permanent support, and Alexandra A. Elbakyan for help with accessing required literature. We are indebted to Natalia Ershova for proofreading. We express our sincere gratitude to the Academic Editor Marcio Pie, Jian-Huan Yang and an anonymous reviewer for their useful suggestions on the earlier version of the manuscript.

### Funding

This work was supported by the Thailand Research Fund (TRF) (DBG6180001) (fieldwork, specimen examination). Molecular experiments, phylogenetic analyses, specimen storage and examination were carried out with the financial support of Russian Science Foundation (RSF grant No. 14-50-00029). The funders had no role in study design, data collection and analysis, decision to publish, or preparation of the manuscript.

### Grant Disclosures

The following grant information was disclosed by the authors:
Thailand Research Fund (TRF): DBG6180001.
Russian Science Foundation: 14-50-00029.

### Competing Interests

The authors declare there are no competing interests.

### Author Contributions

- Parinya Pawangkhanant and Nikolay A. Poyarkov conceived and designed the experiments, performed the experiments, analyzed the data, contributed reagents/materials/analysis tools, prepared figures and/or tables, authored or reviewed drafts of the paper, approved the final draft.
- Tang Van Duong conceived and designed the experiments, performed the experiments, analyzed the data, prepared figures and/or tables, approved the final draft.
- Mali Naiduangchan conceived and designed the experiments, analyzed the data, contributed reagents/materials/analysis tools, authored or reviewed drafts of the paper, approved the final draft, organized and supported fieldwork in Suan Phung District.
- Chatmongkon Suwannapoom conceived and designed the experiments, analyzed the data, contributed reagents/materials/analysis tools, authored or reviewed drafts of the paper, approved the final draft.

### Animal Ethics

The following information was supplied relating to ethical approvals (i.e., approving body and any reference numbers):

No experiments were conducted on living vertebrates. Specimens collection protocols were approved by the Institutional Ethical Committee of Animal Experimentation of the University of Phayao (certificate number UP-AE59-01-04-0022 issued to Chatmongkon Suwannapoom) and strictly complied with the ethical conditions by the Thailand Animal Welfare Act.

### Field Study Permissions

The following information was supplied relating to field study approvals (i.e., approving body and any reference numbers):

Specimens were collected and exported with permission of the the Animal Research Centre, University of Phayao, Thailand, and of the Institute of Animals for Scientific

Purpose Development (IAD), Bangkok, Thailand, under the jurisdiction of the National Research Council of Thailand (NRCT), permit no. U1-01205-2558, issued to Chatmongkon Suwannapoom.

### DNA Deposition

The following information was supplied regarding the deposition of DNA sequences:

Sequences of 16S rRNA genes presented here are accessible via GenBank accession numbers MH581080–MH581082.

### Data Availability

Specimens examined in this study are deposited in herpetological collections of the following museums:

1. School of Agriculture and Natural Resources, University of Phayao (AUP, Phayao, Thailand);

2. Zoological Museum of Moscow University (ZMMU, Moscow, Russia).

### New Species Registration

The following information was supplied regarding the registration of a newly described species:

Publication LSID:

urn:lsid:zoobank.org:pub:766A1EC7-4D17-412D-9DA6-C64E75EA3AFE;

*Leptobrachium tenasserimense* LSID:

urn:lsid:zoobank.org:act:BDDB75C8-EC92-473A-AA81-09D65A18F9D9.

### Supplemental Information

Supplemental information for this article can be found online at http://dx.doi.org/10.7717/peerj.5584#supplemental-information.

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
