# Peer review of "A new species of Leptobrachium (Anura, Megophryidae) from western Thailand"

_PeerJ, doi:10.7717/peerj.5584_

## Round 0.1 · original submission · Minor Revisions

Although both reviews were positive, they identified a number of minor issues that should be addressed before the paper can be accepted. In particular, please pay special attention to the annotated pdf from Reviewer 2.

Reviewer 1 ·

Basic reporting

Some minor comments
Line 47. ‘asian’ should be Asian
Line 64-67 and other places. Please minimize the use of brackets.
Line 67. ‘letter’ should be ‘latter”
Line 168-171. This description is unconventional. Bootstrap values below 50 may indicate that these nodes are not-well-supported; however, most of these nodes are clearly resolved.
Line 205-209. The most meaningful comparison is comparing the distance between L. sp. Vs L. smithi to 1) pairwise distances between recognized sister groups, and 2) the highest intraspecific divergence.
Line 332. “polydominant” should be predominant.
Line 404. Please move this and the following paragraphs to the beginning of the section “comparisons”, because they are the most important.
Lines 434-439. Please remove any sentences that discuss the phylogenetic relationships of the genus. MtDNA may be sufficient to distinguish species, but far from sufficient to build phylogenies.
Line 447-452. Long sentence. Break it.

Table 1 could be an appendix. It is not essential.

Experimental design

No comment

Validity of the findings

Conclusion is well supported

·

Basic reporting

Article is generally well-written, although there are still some minor errors in language. The literature references are sufficient, but the introduction needs more details, particularly on the recognition of subgeneric classification and the “Leptobrachium smithi complex”. Like, the author mentioned “L. smithi complex” in the text a few time, but without any brief background about this species complex. This aspect may be presented in the introduction to make the issue understandable.
The article structure conforms to the acceptable formations.

Experimental design

Materials and methods are correct and sufficient for the scope of the paper. Methods are described with sufficient details are reproducible by other researchers. But the currently presented data is not an integrated taxonomic approach that the author claim in the introduction.

Validity of the findings

The presented data are basically sufficient for supporting the validity of the new species being described, although adding more supporting data on acoustics is highly recommended. Some mistakes in the comparison section, and figure can be noted.

Additional comments

See attached file for more comments/corrections.

---

## Round 0.2 · accepted · Accept

I believe you properly addressed all of the issues raised by the reviewers. Congratulations on finding such a beautiful species!

#